# Knowledge and Competence Regarding the Management of Chronic Kidney Disease among Family Medicine Professionals in the Eastern Province of Saudi Arabia: A Cross-Sectional Study

**DOI:** 10.3390/ijerph21070880

**Published:** 2024-07-06

**Authors:** Abdullah Almaqhawi

**Affiliations:** Department of Family and Community Medicine, College of Medicine, King Faisal University, Al Hofuf P.O. Box 400, Saudi Arabia; aalmuqahwi@kfu.edu.sa

**Keywords:** knowledge, confidence, chronic kidney disease, management, family medicine physician

## Abstract

Diabetes is a significant risk factor for chronic kidney disease (CKD) and a primary cause of global morbidity and mortality, resulting in significant costs to healthcare systems. The management of diabetic CKD in the primary care setting remains an ongoing challenge despite the current best practices in the quality of care. This study evaluated family medicine physicians’ knowledge and confidence regarding managing CKD in the Eastern Province of Saudi Arabia. A self-administered online questionnaire was distributed to physicians through various social media sites and email lists. The largest number of participants reported a full confidence in knowing kidney disease stages, blood pressure targets and the importance of urine albumin–creatinine ratio testing. Overall, 71.8% of physicians reported a high confidence level, followed by 23.9% reporting average and 4.2% reporting low confidence. Being younger and working at PHC were identified as significant predictors of increased confidence. Although most of the physicians reported a high confidence in managing CKD patients, the need for improvement was evident. Age and workplace institutions were the greater contributors to physicians’ confidence. Continuous education among healthcare practitioners is crucial to updating knowledge and providing optimum quality of care among this group of patients.

## 1. Introduction

Globally, diabetes is a major contributor to chronic kidney disease (CKD) [1]. Chronic hyperglycemia brought on by diabetes causes a variety of hemodynamic and metabolic alterations that gradually deteriorate kidney function [2,3]. Diabetes is a major global health concern, with the prevalence expected to increase from 2.8% in 2000 to 4.4% by 2030 across all age groups [4]. According to the International Diabetes Federation, 578 million people will be diagnosed with diabetes worldwide by 2030 [5]. In the context of Saudi Arabia, the country faces a high burden of diabetes, with one study reporting that the age-adjusted prevalence of diabetes was 17.7% [6]. Overall, diabetes is extremely common in Saudi Arabia, with estimates indicating that the country has one of the highest diabetes prevalence rates in the world [7].

Patients with diabetes are at a significant risk for a variety of microvascular and macrovascular complications. Nephropathy, retinopathy and neuropathy are examples of microvascular complications common among people with diabetes that greatly increase the burden of comorbidities [8]. Chronic kidney disease (CKD) is one of the main global causes of morbidity and death [9], and it is usually diagnosed with kidney function tests, kidney imaging and the albumin-to-creatinine ratio [10]. Verma et al. (2021) reported that these complications might result in irreversible harm to the structure and function of tissues, thereby elevating the risk of death and morbidity [11]. Diabetes has been identified as a direct cause of kidney damage and a major contributing factor to the ongoing deterioration of kidney function [12]. The reduced estimated glomerular filtration rate (eGFR) and albuminuria are common in patients with diabetes and nephropathy, which makes renin-angiotensin system blockade therapy necessary [13]. According to studies, diabetes raises the risk of kidney disease progression, which can lead to end-stage kidney disease, a steady decline in eGFR, or even death from cardiovascular or renal causes [14].

The age-standardized prevalence of CKD in Saudi Arabia is estimated to be 9892 per 100,000 people, higher than the estimates for Western Europe (5446 per 100,000 people) and North America (7919 per 100,000 people) [15]. Patients with diabetes in Saudi Arabia have high rates of CKD, ranging from 37.4% to 41.1% [16,17]. This rate reflects a substantial increase over the 5.7% overall prevalence of CKD in the Saudi population [15]. In this population, the use of insulin and oral hypoglycaemic medications is associated with CKD risk factors, along with obesity and hypertension [16]. The Ministry of Health (MOH) in Saudi Arabia approved a “Ten-Year National Executive Plan” for the control of diabetes in response to the burgeoning prevalence of diabetes and its comorbidities in the Kingdom [18]. The plan detailed treatment techniques, prevention strategies and control awareness programs [18].

Primary care physicians (PCPs) are essential in managing diseases, especially long-term conditions such as diabetes [19,20]. PCPs diagnose and treat chronic illnesses with an emphasis on mitigating their effects on patients and lowering the risk of premature death and morbidity [21,22]. Patients with chronic diseases can receive much better care when a disease management strategy is used and the PCP assumes the lead role [19]. Concerning CKD, to prevent complications and manage the symptoms, regular monitoring and coordinated care between primary care and renal services are essential [23]. Successful management requires a team approach involving PCPs, providers and patients, with the degree of participation and interaction adapted to the patient’s stage of CKD [24]. Studies conducted in a variety of countries have all highlighted the importance of increasing family medicine physicians’ understanding of the diagnosis and treatment of CKD. For instance, Godswill (2016) reported that Nigerian family medicine trainees lacked sufficient knowledge in this domain, suggesting the necessity for enhanced instruction [25]. Further, although recent evidence suggested that family physicians in Poland have a reasonable understanding of the causes, risk factors and progression of CKD, additional education and factual knowledge are still needed in this profession [26]. Research has also shown that doctors’ understanding of the definition of CKD is even poorer in West Africa 38.5% [27], and only 38% of physicians from Pakistan were aware that eGFR could be used to diagnose CKD [28].

This is the first study to assess family medicine doctors’ confidence and knowledge about treating CKD in the Eastern Province of Saudi Arabia. This endeavor could yield valuable information for formulating a plan of action to promote effective CKD management among family medicine physicians.

## 2. Materials and Methods

### 2.1. Study Design and Participant Population

This cross-sectional study was conducted among family medicine physicians employed in primary care settings in the Eastern Province of Saudi Arabia using a survey we created in Google Forms. Participants are recruited from various clusters across different Eastern Province areas to capture a diverse demographic and geographic representation. It announced an anonymous online survey on social media platforms, including official WhatsApp for each cluster, to increase response rates. Additionally, the questionnaire was disseminated via email through the various family medicine clusters’ mailing databases, targeting approximately 240 physicians. The clusters involved were al-Ahsa and Eastern Health Cluster, Dammam, Al Khobar, Dhahran, and Qatif. Out of the approximately 240 doctors contacted, 71 responded and participated in the study. While this represents a response rate of approximately 30%, it is important to consider the diversity within this sample in terms of geographic distribution and professional backgrounds. Distribution of the survey took place between 15 December 2023, to 16 April 2024. Every participant in the study provided informed consent before beginning the questionnaire. In addition, they read information about the aim of the study, and the survey terminated automatically if they disagreed. The questionnaire used to collect data was validated and adapted from a previously published peer-reviewed article [29].

The primary outcome of this study was to evaluate family medicine physicians’ knowledge and confidence regarding managing CKD in the Eastern Province of Saudi Arabia. The recruitment began after receiving ethical approval from the ethical committee of the College of Medicine at King Faisal University. 

### 2.2. Questionnaire Criteria

Family medicine professionals’ knowledge and competence regarding the management of CKD were assessed using a 12-item questionnaire scored on a 5-point Likert scale ranging from 1 (“not confident about this subject”) to 5 (“fully confident in this area and could teach others”). The total confidence score was calculated by summing all 12 items [26]. Possible scores ranged from 12 to 60 points, with higher scores indicating greater confidence in managing CKD. We categorized scores into three confidence levels: <50% indicated poor confidence, 50% to 75% indicated moderate confidence, and above 75% indicated good confidence [30].

### 2.3. Statistical Analysis

Categorical variables were reported as frequencies and proportions (%), whereas continuous variables were computed and expressed as means and standard deviations. The association between confidence and the socio-demographic characteristics of the physicians was evaluated using the Mann–Whitney Z-test and the Kruskal–Wallis H-test. The normality test was evaluated using the Shapiro–Wilk test and the Kolmogorov–Smirnov test. Since the confidence scores followed a non-normal distribution, non-parametric tests were applied. A *p*-value of less than 0.05 was considered statistically significant. All statistical data were analyzed using Statistical Packages for Social Sciences version 26 (IBM Corp., Armonk, NY, USA).

## 3. Results

This study enrolled 71 family medicine professionals. As seen in Table 1, 69% were aged between 31 and 40 years, with females comprising more than half (56.3%) of the total sample. Almost all (98.6%) of the physicians were Saudi nationals, with most working in MOH (60.5%) or as consultants (56.3%). In terms of location, the largest group (42.3%) of physicians was practicing in Al Ahsa. 

When examining the confidence of the family physicians regarding the management of CKD (Appendix A), we examined family physicians who expressed full confidence in understanding the significance and importance and the ability to interpret the urine albumin–creatinine ratio testing in individuals living with diabetes and those who expressed the ability to interpret urine (56.3% and 46.5%, respectively). More than half (59.2%) of the physicians expressed full confidence in their knowledge of the stages of kidney disease, and just under half (47.9%) expressed full confidence in their knowledge of the criteria for the diagnosis of CKD and diabetic kidney disease. In terms of treatment, 38% of physicians indicated full confidence in their knowledge of the next steps in terms of treatment after diagnosis, but only 26.8% expressed full confidence in their understanding of how to predict CKD prognosis using albuminuria and estimated GFR categories. Just over one third (36.6%) of participants reported full confidence in their ability to recognize the possible signs and symptoms of more advanced CKD. Less than half (46.5%) of the physicians expressed full confidence in their awareness of kidney disease as a risk multiplier for CVD and other complications, and less than one third (31%) reported full confidence in their ability to select appropriate management to prevent or slow the progression of CKD. However, 57.7% of the respondents felt fully confident in their ability to understand blood pressure targets, and 54.9% were confident in their understanding of the use of treatments, including angiotensin-converting enzyme inhibitors or angiotensin II receptor blockers. In addition, 50.7% expressed full confidence in initiating diabetes medications that have particular benefits in diabetic kidney disease. Based on the confidence items, the overall mean confidence score for the sample was 49.5 (SD 9.66), with 71.8%, 23.9% and 4.2% of the sample reporting high, average and low confidence levels, respectively.

When analyzing the association between confidence scores and the socio-demographic characteristics of the family medicine physicians (Table 2), our results indicated that a higher confidence score was associated with being younger (Z = 2.022; *p* = 0.043) and working in PHC (H = 9.465; *p* = 0.024). No significant differences were observed among confidence scores based on gender, professional degree, or city of practice (*p* > 0.05).

## 4. Discussion

This study evaluated the knowledge and competence of PCPs in managing CKD patients. According to our results, physicians demonstrated high confidence levels. Based on the 12 knowledge and confidence items, the overall mean confidence score was 49.5 out of 60 possible points, with a majority of PCPs categorized as having a high level of confidence (71.8%), followed by average (23.9%) or low (4.2%) levels. To our knowledge, this is the first study conducted in Saudi Arabia to assess the overall confidence level of PCPs regarding CKD management. Apart from one study conducted in the United Kingdom, which generalized confidence levels of PCPs in all aspects, no prior research has been conducted using similar criteria on the same subject [29]. However, a few studies have assessed PCPs’ knowledge of CKD with mixed results. For instance, Jazienicka-Kiełb et al. (2022) documented a reasonably high level of knowledge regarding the risk factors, causes and course of CKD, while Agaba et al. (2012) reported a lack of understanding of the CKD management guidelines among non-nephrology specialist physicians [26,29]. Thus, this study is an important contribution to the literature given the difficulties in managing this disease, as gauging the competency levels of PCPs is vital for determining training needs and providing optimum care. 

Being younger was the only significant factor associated with increased confidence. A previous report published in the United States comparing the knowledge of PCP subspecialties revealed that the PCPs with a subspecialty of internal medicine had more than 3-fold higher odds of showing satisfactory knowledge levels than family practice specialists [31]. However, a Nigerian study comparing family physicians and non-nephrology internists found no significant differences between their levels of knowledge (*p* > 0.05) [27]. In our study, we found that the gender, professional degree and practice city had no significant effect on practicing physicians’ confidence levels. Our findings corroborate those of Choukem et al. (2016), who reported that knowledge of CKD did not significantly differ based on the physician’s gender, level of training, hospital practice, or city of practice (*p* > 0.05) [32].

When examining the details of confidence toward CKD management, we noted the least amount of confidence in the specific criteria for treating and managing CKD patients. Most notably, the gaps were seen in the knowledge of the appropriate guidelines to predict a CKD prognosis, with 18.3% and 4.2% reporting a lack of confidence and needing more knowledge. Similarly, 22.5% and 7% demonstrated confidence, but with support and requiring more training about the next steps in post-diagnosis treatment. Our results are consistent with those of Seidu et al. (2023), who found that PCPs’ ratings ranged from 16.5% to 21.8% when asked if they felt fully confident in CKD presentation, prognosis and staging [29]. Contradicting these reports, Wolide et al. (2020) indicated that most care providers knew the CKD stages and their risk factors, many were interested in future CKD management training (71.8%), and most tended to refer CKD patients to a senior physician or nephrologist (78.5%) [33]. 

Moreover, a study published in Brazil suggested that isolated serum creatinine was the most widely used test for early diagnosis of CKD, while health appointments and drug intervention were the prominent disease-prevention strategies [34]. This finding is consistent with a study conducted in Pakistan (Yaqub et al., 2013), which reported that serum creatinine was the first choice for estimating kidney function (78.1%) and 24 h collection of urine the first one for clearance of creatinine (63.8%) [28]. However, only 37.9% used mathematical formulas for the estimation of eGFR. Among internal medicine residents in the United States, the selected interventions for slowing the progression of CKD included aggressive glycaemic control, lipid control, dietary salt restriction, weight loss, the use of angiotensin-converting enzyme (ACE) inhibitors or angiotensin receptor blockers, and smoking, with many of the residents reporting they knew about using eGFR to estimate CKD progression [32]. In our study, most PCPs were either confident without support or fully confident about the importance of using and their ability to interpret the urine albumin–creatinine ratio. In addition, their confidence was high in their knowledge of the next steps for appropriate treatment, and they felt adept at selecting proper treatment and management to prevent the progression of CKD, including the beneficial effect of SGLT2-Is and glucagon-like peptide 1 receptor agonists. 

PCPs reported confidence in determining the stages of CKD, the signs and symptoms, and the association between kidney disease and CVD, with 70.4% to 86% of the physicians reporting confidence without support to complete confidence. In Madinah, approximately two thirds of physicians sampled were aware of the five stages of CKD, while only 16% recognized CKD patients with Stage 4 as requiring a referral to a specialist, such as a nephrologist [35]. In Cameroon, the majority of surveyed physicians were aware of the major risk factors of CKD, including diabetes and hypertension, and were confident in recognizing CKD complications, such as anaemia, hypertension, uraemia and hyperkalaemia [32]. However, in the same study, only 12.7% of the physicians reported that they would use serum creatinine alone for CKD diagnosis, and only 21.9% would refer patients at a late stage. In our study, most of the PCPs (88.7%) reported that they were confident without support and/or fully confident in their knowledge of blood pressure targets. This is consistent with the literature, indicating that physicians were adept at the target goal of controlling blood pressure [28,31,36,37].

Concerning barriers to CKD management, Al-Zaman et al. (2023) stated that although most physicians reported encountering few barriers to CKD management, general practitioners working in PHCs reported experiencing substantial barriers, and female physicians reported significantly more barriers than male physicians [35]. Sperati et al. (2019) cited that barriers to managing CKD primary care varied based on patients’ awareness of CKD, poor adherence to treatment recommendations, providers staying with current CKD guidelines, and healthcare systems’ inflexible electronic medical records and limited time and resources [38]. In our study, we did not explore barriers to managing CKD; however, barriers should be considered in future research. The sample size and response rate limitations are acknowledged in our manuscript. Nevertheless, we also emphasize the measures implemented to guarantee that the sample, in spite of its size, is representative of the larger physician population in the targeted areas. In our study, we did not explore barriers to managing CKD. Also, longer recruitment times and additional engagement techniques should be implemented in order to boost response rates and strengthen the sample’s resilience.

## 5. Conclusions

PCPs in our study demonstrated a high confidence in managing patients with CKD. Higher reported levels of confidence were seen more frequently among younger physicians practicing in primary healthcare. Although the overall confidence of PCPs was good, some PCPs expressed a lack of confidence, particularly regarding knowledge in predicting CKD prognosis, the recognition of the possible signs and symptoms of more advanced CKD and the subsequent process of the treatment procedures after diagnosis. Hence, PCPs should update their CKD knowledge to provide a thorough and efficient method of managing CKD patients in primary care. The implementation of educational programs based on the presented research findings can enhance the practical value of the manuscript.

## Figures and Tables

**Table 1 ijerph-21-00880-t001:** Socio-demographic characteristics of family medicine physicians (N = 71).

Study Variables	*n* (%)
Age group	
22–30 years	07 (09.9%)
31–40 years	49 (69.0%)
>40 years	15 (21.1%)
Gender	
Male	31 (43.7%)
Female	40 (56.3%)
Nationality	
Saudi	70 (98.6%)
Non-Saudi	01 (01.4%)
Workplace institution	
MOH	43 (60.5%)
MNGHA	02 (02.8%)
IAU	06 (08.5%)
KFMMC	03 (04.2%)
KFU	06 (08.5%)
Private hospital	11 (15.5%)
Professional Degree	
Specialist	27 (38.0%)
Fellow	04 (05.6%)
Consultant	40 (56.3%)
City of practice	
Al Ahsa	30 (42.3%)
Qatif	14 (19.7%)
Dammam	16 (22.5%)
Khobar	07 (09.9%)
Dahran	04 (05.6%)

MOH; Ministry of Health. MNGHA; Ministry of National Guard Health Affairs. IAU; Imam Abdulrahman Bin Faisal University. KFMMC; King Fahad Military Medical City. KFU; King Faisal University.

**Table 2 ijerph-21-00880-t002:** Association between confidence scores and socio-demographic characteristics of family medicine professionals (N = 71).

Factor	Confidence Score (60) Mean ± SD	Z/H-Test	*p*-Value
Age group ^a^			
≤35 years	52.5 ± 7.92	2.022	**0.043 ****
>35 years	48.0 ± 10.2		
Gender ^a^			
Male	49.0 ± 11.4	0.122	0.903
Female	49.9 ± 8.23		
Workplace institution			
MOH	49.1 ± 10.5	0.218	0.827
Non-MOH	50.2 ± 8.45		
Professional degree ^a^			
Specialist/Fellow	50.2 ± 7.83	0.093	0.926
Consultant	49.1 ± 10.9		
City of Practice ^a^			
Inside Al Ahsa	48.6 ± 10.9	0.607	0.544
Outside Al Ahsa	50.3 ± 8.76		

^a^ *p*-value calculated using the Mann–Whitney Z-test. ** *p* < 0.05.

## Data Availability

The datasets used and analyzed during the current study are available from the corresponding author on reasonable request.

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
