# Peer review of "Knowledge and Competence Regarding the Management of Chronic Kidney Disease among Family Medicine Professionals in the Eastern Province of Saudi Arabia: A Cross-Sectional Study"

_ijerph, 2024, doi:10.3390/ijerph21070880_

Round 1

Reviewer 1 Report

Comments and Suggestions for Authors

Revision on the manuscript entitled "Knowledge and Competence Regarding the Management of Chronic Kidney Disease Among Family Medicine Professionals in the Eastern Province of Saudi Arabia: A Cross-Sectional Study” submitted in the Journal of Environmental Research and Public Health. This article sought to evaluate family medicine physician’s knowledge and confidence about managing CKD in the eastern province of Saudi Arabia. It is a cross-sectional study that analyzed 71 physicians through questionnaires about chronic kidney disease management-related topics. The study highlights the importance of continuous education among healthcare practitioners to attain better management in CKD care. The manuscript is well-structured and addresses an important aspect of healthcare. However, the focus on diabetes, while clinically relevant, seems to overshadow the main aim related to CKD management. The manuscript offers valuable insights but would benefit from a more focused examination of CKD independent of the extensive discussion on diabetes. The revisions suggested should help to align the content more closely with the study’s objectives. Find below my comments:

Major

The manuscript extensively discusses diabetes, its complications, and prevalence. While the connection between diabetes and CKD is clinically significant, the emphasis on diabetes may detract from the primary focus of the study, which is to assess confidence and knowledge specifically related to CKD management. I recommend the authors reassess the content balance to ensure that CKD management is clearly the focal point of the paper.

It would be beneficial for the manuscript to clarify in the introduction why diabetes is given significant attention and how this directly relates to the primary objectives of the study. If diabetes is intended to be a major focus, then the aims of the study should be adjusted accordingly to reflect this. Or change the text to be more adjusted to your objectives.

The manuscript should provide more details on the sampling strategy used to ensure the representativeness of the sample and enhance the credibility of the study findings.

Further details on the development and validation of the questionnaire would strengthen the study's methodology section. If the tool is new, descriptions of its pilot testing and validation process should be included.

The manuscript should include explicit statements confirming that all statistical assumptions for the non-parametric tests were checked and satisfied.

A broader discussion comparing these findings with international studies could provide more context and highlight the study’s contributions more effectively.

Further details about the ethical considerations, particularly concerning the online collection of data and ensuring participant confidentiality, would be pertinent.

Specific recommendations for educational programs or policy changes based on the findings would add practical value to the manuscript.

Author Response

I recommend the authors reassess the content balance to ensure that CKD management is clearly the focal point of the paper.

I add more about CKD according to the reviewer's comment.

It would be beneficial for the manuscript to clarify in the introduction why diabetes is given significant attention and how this directly relates to the primary objectives of the study.

I try to link it to diabetes, as it is very common in Saudi Arabia as well as to the family medicine physician.

If diabetes is intended to be a major focus, then the aims of the study should be adjusted accordingly to reflect this. Or change the text to be more adjusted to your objectives.

The aim of the study to evaluate family medicine doctors' confidence and knowledge about treating CKD

The manuscript should provide more details on the sampling strategy used to ensure the representativeness of the sample and enhance the credibility of the study findings.

I modified it according to the reviewer's comment.

Further details on the development and validation of the questionnaire would strengthen the study's methodology section. If the tool is new, descriptions of its pilot testing and validation process should be included. ( it was taken from the previous study also in English )

I modified it according to the reviewer's comment. It was in English as the participants were physician

The manuscript should include explicit statements confirming that all statistical assumptions for the non-parametric tests were checked and satisfied.

it is stated that the confidence score follow the non-normal distribution, so the non-parametric test were applied in the Statistical analysis.

A broader discussion comparing these findings with international studies could provide more context and highlight the study’s contributions more effectively.

 I modified according to the reviewer's comment

Further details about the ethical considerations, particularly concerning the online collection of data and ensuring participant confidentiality, would be pertinent.

Specific recommendations for educational programs or policy changes based on the findings would add practical value to the manuscript.

I explained that more in the methodology part. Also, I modified it according to the reviewer's comment.

Reviewer 2 Report

Comments and Suggestions for Authors

Thank you for the opportunity to review this manuscript. I have a number of concerns and suggestions for the Author to consider:

1) The abstract is a little repetitive, please make sure that concepts are expressed only once.

2) This looks like a non-representative survey (or at least, nothing is said to have been done to ensure representativeness), so the Author needs to discuss about proneness to selection bias and how this may have impacted on the results . Also, completeness appears to be an issue.

3) Was the questionnaire validated? Is it the translation of a validated questionnaire? Or is it a completely self-made questionnaire? If so, this is a main limitation that needs to be acknolwedged.

4) Why are mean reported if the distributions are non normal and non-parametric tests are used?

5) The Shapiro-Wilk and Kolmogorov-Smirnov test are for normality, not for collinearity, if I'm not mistaken.

6) How is it possible that over 60% of general practitioners work at the MOH? Which kind of GPs work at the MOH? This raises further concerns regarding representativeness and selection bias. It looks like the participants are a heavily selected subset of GPs...

7) Table 2 should be either transformed into a graphic (e.g. histogram) or moved into supplementary material. It is too big and difficult to make quick sense of right now.

8) What is PHC, and why is it not reported in Table 3?

9) The conclusion that "physicians demonstrated high confidence levels" is way too simplistic and actually quite misleading. 30% of GPs do not show high confidence, and this is a large proportion, considering that this medical condition is very common (all GPs should be confident in managing diabetes and chronic kidney disease!). 

10) The discussion lacks considerations on recommendations, implications, etc. Why did the author conduct this survey? Knowledge that does not shape future actions is useless in this field, so some recommendations should be made (compulsorily).

Comments on the Quality of English Language

Some revision required, some sentences are not built correctly.

Author Response

1) The abstract is a little repetitive, please make sure that concepts are expressed only once.

Thank you for your feedback, I modified it according to the reviewer's comment.

2) This looks like a non-representative survey (or at least, nothing is said to have been done to ensure representativeness), so the Author needs to discuss about proneness to selection bias and how this may have impacted on the results . Also, completeness appears to be an issue.

    • The target population consisted of doctors from cluster in Al-Ahsa and Dammam regions.
    • An email was sent to all cluster in Al-Ahsa and Dammam, targeting approximately 240 doctors.
    • We utilized official social media channels, specifically WhatsApp, to disseminate information and invitations to the doctors' gatherings in both Al-Ahsa and Dammam. Also, to boost the response rates  we sent the link to the following social media platforms
    • Out of the approximately 240 doctors contacted, 71 responded and participated in the study. While this represents a response rate of approximately 30%, it is important to consider the diversity within this sample in terms of geographic distribution and professional backgrounds.
    • Follow-up messages and reminders were sent to maximize participation. Additionally, efforts were made to engage with doctors and address any queries they had about the study, thereby encouraging their involvement.
    • We acknowledge the limitation of the sample size and response rate in our manuscript. However, we also highlight the steps taken to ensure that the sample, despite its size, is representative of the broader population of doctors in the targeted regions.
    • We recommend future studies to consider extended recruitment periods and additional engagement strategies to improve response rates and enhance the robustness of the sample.

3) Was the questionnaire validated? Is it the translation of a validated questionnaire? Or is it a completely self-made questionnaire? If so, this is a main limitation that needs to be acknolwedged.

I modified it according to the reviewer's comment.

4) Why are mean reported if the distributions are non normal and non-parametric tests are used?

I used z-test and H-test to indicate the distribution of the mean, U-test indicates median. I used mean for more clarity with interpretation

5) The Shapiro-Wilk and Kolmogorov-Smirnov test are for normality, not for collinearity, if I'm not mistaken.

I modified it according to the reviewer's comment.

6) How is it possible that over 60% of general practitioners work at the MOH? Which kind of GPs work at the MOH? This raises further concerns regarding representativeness and selection bias. It looks like the participants are a heavily selected subset of GPs...

Yes must likely the PHC belong to MOH.

7) Table 2 should be either transformed into a graphic (e.g. histogram) or moved into supplementary material. It is too big and difficult to make quick sense of right now.

I modified it according to the reviewer's comment.

8) What is PHC, and why is it not reported in Table 3?

This study is aim to assess the family medicine physicians' knowledge and confidence regarding managing CKD who work in PHC

9) The conclusion that "physicians demonstrated high confidence levels" is way too simplistic and actually quite misleading. 30% of GPs do not show high confidence, and this is a large proportion, considering that this medical condition is very common (all GPs should be confident in managing diabetes and chronic kidney disease!). 

This is reflect their confidence level dealing with CKD cases. Which will be a good to know and work to raise that. Family medicine physicians are primary care physicians (PCPs) who care for patients as whole people, provides comprehensive medical care to patients of all genders.

10) The discussion lacks considerations on recommendations, implications, etc. Why did the author conduct this survey? Knowledge that does not shape future actions is useless in this field, so some recommendations should be made (compulsorily).

I modified it according to the reviewer's comment.

Round 2

Reviewer 2 Report

Comments and Suggestions for Authors

Thanks for improving the manuscript.